# Association between 'Emergency obstetric & newborn care readiness' and delivery service utilization in Bangladesh: Evidence from national health facility assessment surveys

**Kaji Keya-Korotki**[1], **Nabil Natafgi**[2*], **Mark Macauda**[3], **Syed Abdul Hamid**[4], **M. Mahmud Khan**[5]

**1** Department of Public Health State of South Carolina, Columbia, South Carolina, United States of America, **2** Department of Health Services Policy and Management, Arnold School of Public Health, University of South Carolina, Columbia, South Carolina, United States of America, **3** Colorectal Cancer Prevention Network, University of South Carolina, Columbia, South Carolina, **4** Institute of Health Economics, University of Dhaka, Dhaka, Bangladesh, **5** Department of Health Policy and Management, College of Public Health, University of Georgia, Athens, Gerogia, United States of America

\* nnatafgi@mailbox.sc.edu

## Abstract

### Background

Bangladesh faces one of the highest global burdens of maternal and newborn deaths, primarily caused by hemorrhage and eclampsia/preeclampsia. Despite reducing maternal mortality by 40%, comprehensive medical interventions remain insufficient in health facilities.

### Objectives

This study assessed the readiness of sub-district hospitals in Bangladesh to provide Emergency Obstetric and Newborn Care (EmONC) and analyzed the association between facility readiness and delivery rates.

### Methods

Using Health Facility Assessment Survey data from 140 hospitals in 2014 and 141 in 2017, facility readiness was measured based on nine signal functions: administering antibiotics, oxytocin, and anticonvulsants; providing blood transfusions; performing cesarean and assisted vaginal deliveries; managing retained placentas and products of conception; and neonatal resuscitation. Donabedian's model guided the analysis. Multiple linear regression examined associations between facility readiness and delivery rates using 2017 data.

### Results

Between 2014 and 2017, the availability of signal functions such as oxytocin (85% to 95%), anticonvulsants (58% to 63%), and blood transfusions (22% to 38%) improved. In 2014, 77% of facilities had at least 5 signal functions, 49% had 7, and 6% had all 9. By 2017,

**Data availability statement:** The dataset related to the manuscript are uploaded here: Keya-Korotki, Kaji. Plos One manuscript data. Ann Arbor, MI: Inter-university Consortium for Political and Social Research [distributor],

2025-02-12. https://doi.org/10.3886/E219173V1

**Funding:** The author(s) received no specific funding for this work.

**Competing interests:** The authors have declared that no competing interests exist.

these increased to 83%, 56%, and 8%, respectively. Despite these improvements, the mean readiness index remained nearly unchanged (0.67 in 2014 vs. 0.69 in 2017). Only 8% of facilities performed fewer than 52 deliveries annually in 2017, while 26% conducted over 500. Regression analysis revealed a significant association between readiness scores and delivery rates (p = 0.009).

## Conclusion

While certain indicators improved, overall readiness stagnated due to shortages of anesthetists, gynecologists, and essential supplies. With 64% of surveyed sub-district hospitals classified as comprehensive care facilities, resource and staffing investments are crucial to enhance readiness and reduce maternal and newborn mortality.

## Introduction

Maternal mortality remains a significant global challenge, with hemorrhage, hypertensive disorders, and sepsis accounting for over half of maternal deaths [1]. While South Asian countries, including Bangladesh, have made remarkable progress in reducing maternal mortality, Bangladesh's maternal mortality ratio (MMR) remains higher than neighboring countries [2]. Over the past decade, Bangladesh has achieved a 40% reduction in its MMR [3]. However, this progress is constrained by continued reliance on home deliveries, often conducted without the assistance of trained midwives or nurses.

To address this, Bangladesh implemented policies promoting home deliveries by skilled midwives over the past 20 years [4,5]. This approach has proven effective for uncomplicated deliveries, reducing maternal deaths. However, complications such as hemorrhage and eclampsia—responsible for around 54% of maternal deaths in Bangladesh—require advanced interventions available only in health facilities equipped with appropriate medicines, supplies, and trained medical professionals [6]. For instance, managing hemorrhage requires the prompt administration of parenteral oxytocin, manual removal of the placenta, removal of retained products, and provision of blood transfusions. Similarly, managing eclampsia necessitates the use of parenteral anticonvulsants. Consequently, improving facility readiness is essential to address these challenges and further reduce maternal mortality.

In addition to promoting skilled midwifery, Bangladesh has strengthened its public hospital network by upgrading sub-district-level hospitals from basic to comprehensive Emergency Obstetric and Newborn Care (EmONC) facilities. Basic EmONC hospitals typically have 31 beds, while comprehensive EmONC hospitals have 50 beds and can provide cesarean sections and blood transfusions [7]. Of the 420 sub-district hospitals in Bangladesh, 297 have been upgraded to comprehensive EmONC facilities. These hospitals serve populations ranging from 250,000 to 400,000 [7]. However, persistent issues, including shortages of logistics, supplies, medicines, and the absence of gynecologist-anesthetist pairs, continue to delay the care of complicated deliveries, increasing the risk to maternal and neonatal lives. These challenges highlight the need to assess facility readiness and its association with facility-based deliveries.

Studies demonstrate a strong association between facility-based deliveries and neonatal survival in 67 low-income countries [8,9]. Many low-income countries encourage women to deliver at the nearest health facility equipped for childbirth. Yet, inadequate quality of care contributes to maternal and neonatal deaths, often occurring either at facilities or during transfers to better-equipped centers[10,11]. Ensuring consistent quality of care remains a challenge in regions like South Asia and sub-Saharan Africa, where health systems are often under-resourced and non-responsive [12].

Research on the quality of obstetric care in low- and middle-income countries (LMICs) highlights critical gaps, particularly in lower-level facilities [13–15]. Although associations between delivery rates and quality outcomes have been established in high-income countries, such studies are limited in LMICs. For example, U.S. hospitals with fewer than 25 deliveries per month have 50% higher adverse maternal outcomes compared to hospitals with higher delivery rates [16,17].

In Bangladesh, systematic research linking facility readiness and delivery rates remains scarce. While Kruk et al. [15] found that facilities with cesarean section capacity and higher delivery volumes scored better on maternal care quality in five African countries, similar studies at sub-district-level hospitals in Bangladesh are lacking. To address this gap, this study aims to (1) assess the readiness of sub-district hospitals to provide EmONC services, (2) evaluate changes in facility readiness between 2014 and 2017, and (3) examine the association between facility readiness and delivery service utilization at sub-district hospitals in Bangladesh.

## Method

### Public health care structure in Bangladesh

The country's healthcare structure spans six levels: national, divisional, district, sub-district (upazila), union, and ward. Facility-based childbirth services are offered at the sub-district level and district, divisional, and national level through Upazila Health Complexes (UHCs). Of the 420 UHCs, 297 are 50-bed hospitals capable of providing cesarean delivery. These hospitals, located at the community level, also offer antenatal care, postnatal care, abortion services, and care for maternal health complications. Fifty-bed UHCs represent upgraded versions of 31-bed facilities and are the first level of hospitals equipped with gynecologist-anesthetist pairs and blood transfusion systems.

### Settings and type of facilities

This manuscript studied the overall quality of sub-district hospitals with an emphasis on the facility readiness of comprehensive delivery care. All of them provide free delivery care to the clients. Staff, logistics, medicines, and all other expenses are paid and maintained with tax money by the government. A basic EmONC facility should have seven signal functions, while a comprehensive EmONC should have nine signal functions (Table 1).

### Signal function (SF)

Some medical interventions termed 'signal functions' defined by the United Nations (UN) could be used for preventing maternal and neonatal deaths [18, 19]. EmONC is categorized into two levels based on the services provided: Basic EmONC and Comprehensive EmONC. The **Basic EmONC includes** facilities capable of performing the following seven signal functions: (i) Administering parenteral antibiotics; (ii) Administering uterotonic drugs (e.g., oxytocin) for active management of labor; (iii) Administering parenteral anticonvulsants (e.g., magnesium sulfate) for eclampsia or pre-eclampsia; (iv) Manual removal of the placenta; (v) Removal of retained products of conception (e.g., through manual vacuum aspiration); (vi) Performing assisted vaginal delivery (e.g., with forceps or vacuum extraction); and (vii) Basic neonatal resuscitation (e.g., using a bag and mask). **Comprehensive EmONC** reflects facilities providing the above seven functions, plus: (viii) Performing cesarean sections and (ix) Administering blood transfusions [19–21].

### Data sources

Two datasets about childbirth and quality of care were used from the most recent Service Provision Assessment (SPA) in 2014 and 2017. Those two datasets were the two most recent

**Table 1. Characteristics and Standards of Basic and Comprehensive Emergency Obstetric and Newborn Care (EmONC) Facilities in Bangladesh.**

| EmONC standard indicators | Basic EmoNC facility | Comprehensive EmONC Facility |
|---|---|---|
| Population served | 250,000-400,000 | 250,000-400,000 |
| Availability per sub-district | 1 | 1 |
| Signal functions | 7 | 9 |
| Health care workers | Midwives or nurses trained in midwifery | Obstetrician, anesthetist, midwives or nurses trained in midwifery |
| Supplies and pharmaceuticals | Basic obstetric drugs (e.g., oxytocin, magnesium sulfate), basic instruments for vaginal delivery | Full set of obstetric drugs, equipment for cesarean sections, and blood transfusion supplies |
| Infrastructure | 31 beds | 50 beds |
| # of facilities surveyed in 2014 (SPA) | Not available | Not available |
| # of facilities surveyed in 2017 (SPA) | 46 basic facilities | 86 comprehensive facilities |

available surveys at the time of analysis (and writing of this manuscript). Due to the COVID-19 pandemic, the survey scheduled for 2020 was postponed, making the 2017 dataset the latest available for analysis. The 2014 SPA survey had 140 sub-district hospitals, and the 2017 SPA had 141 hospitals. The first survey did not have any variables to stratify the hospitals as basic and comprehensive EmONC. The 2017 SPA had 86 facilities offering comprehensive EmONC (50 beds), 36 hospitals had basic EmONC, and three hospitals had 10 beds with basic EmONC. Population standardized delivery rate and service utilization data of 2017 were used to study the association between the quality of delivery care and delivery service utilization rate.

a. _SPA survey:_ The facilities surveyed in 2014 and 2017 are not identical cohorts. Some overlap exists, but the surveys were conducted cross-sectionally at different time points, with variations in included facilities. The surveys were conducted in districts, sub-districts, unions, NGOs, and private facilities. The 2017 SPA survey, initially scheduled for 2017, was delayed due to logistical issues and was conducted in 2019 (a delay not uncommon in low-income countries). In this article, we studied a total of 140 sub-district facilities surveyed in 2014 and 141 facilities surveyed in 2017, and those facilities were included in the analysis. A brief description of both surveys is available from the facility assessment report [4,22].

b. _Service utilization dataset:_ Delivery rate and service utilization data of the respected facilities were obtained from the yearly health bulletins published by the Ministry of Health and Family Welfare, Government of Bangladesh. The data is publicly available on the Directorate General of Health Service's website [23]. Service utilization data has been publicly available since 2016 and included in this study for 2017. Out of 141 facilities in 2017, a total of 7 facilities had missing information and finally, 134 facilities (141-7 = 134 facilities) were included in the service utilization analysis.

## Theoretical model

This study is based on Donabedian's theoretical model of quality of care, which provides a comprehensive framework for assessing **structure**, **process**, and **outcomes** [24]. In the context

of this study, quality of care is operationalized as **facility readiness**, with the following components: **Structure**: Refers to the availability of logistics, supplies, equipment, and personnel (e.g., midwives, obstetricians, anesthetists). **Process**: Denotes the activities involved in providing care, such as performing the nine signal functions for Emergency Obstetric and Newborn Care (EmONC), including diagnosis, prescription, and the execution of critical interventions. **Outcome**: Refers to the results of care delivery, such as rates of facility-based deliveries, cesarean sections, and maternal or neonatal health outcomes.

In this study, the **process** dimension was emphasized by focusing on the availability and performance of signal functions as a proxy for the quality of care provided. While this emphasis may appear to diverge from Donabedian's comprehensive framework, it is a practical approach in low- and middle-income countries (LMICs), where reliable data on outcomes and structural readiness are often limited. The focus on process indicators provides actionable insights into system-level weaknesses that are amenable to improvement within resource-constrained settings[25].

The Donabedian model was chosen for its adaptability and relevance to health system assessments in LMICs. By examining signal functions, this study identifies gaps in facility readiness and highlights areas where improvements in structure (e.g., increasing supplies and personnel) could strengthen processes and, ultimately, outcomes. This alignment ensures the model remains a guiding framework for understanding and addressing quality gaps in maternal and newborn care within Bangladesh.

## Study design and statistical analysis

The study calculated the facility readiness of the sub-district hospitals both for the availability of the care – defined as the facility ever performed that care, and for the preparedness of the care – defined as the facility performed that care in the previous 3 months. We have also studied some key practices, such as maternal and neonatal death review and the practice of kangaroo mother care, given that these practices are important indicators of quality improvement effort.

For statistical analysis, univariate methods were used to calculate frequencies, means, standard deviations, and proportions to describe facility readiness and associated practices. Bivariate analyses were conducted using chi-square tests to examine categorical variables and independent t-tests to compare means between groups where applicable. Cross-tabulations were performed to explore associations between facility characteristics (e.g., readiness and availability of services) and key outcomes.

For multivariate analysis, two multiple linear regression models were employed to investigate the association between facility readiness and outcomes. The first model used population-standardized delivery rates as the dependent variable, while the second model examined cesarean delivery rates. Significance was determined at a p-value of $< 0.05$ for all statistical tests.

## Study variables: Dependent variables

**Delivery rate.** The total delivery of each study facility was calculated by summing up all different kinds of delivery (number of normal, assisted, and C-section deliveries) in the study hospital in 2017. The number of total deliveries was converted into rates by dividing by the population of the sub-district and multiplying by 100,000.

**Delivery service utilization.** The delivery rate was categorized into a few thresholds: less than 52, between 53-183, 184-365, 366-500, and 501 + per year. These thresholds were selected for interpretability (52 is one delivery per week, 183 is one every other day, and 366 is one per

day) and to reflect international thresholds (e.g., 500 births per year) in the USA and UK, and provide a roughly balanced distribution of both facilities and births per category [15].

**C-section delivery rate.** Total number of C-section deliveries conducted at each UHC.

The number of c-section deliveries was converted into rates by dividing by the sub-district's population and multiplying by 100,000.

## Study variables: Independent Variables

**Facility Readiness Index.** Facility readiness was measured as the availability of signal function indicators in the facility and the practice of signal function indicators in the past 3 months. The mean or average of these nine indicators was the signal function quality score (Facility Readiness Index). In the signal function quality index, all nine indicators are process indicators, as these indicators take place during diagnosis, prescription, and treatment. The index was compiled by summing up all nine available indexes and dividing by nine. For example, if a facility has six indicators available out of the nine indicators, the index was calculated as six divided by nine, making it 0.66. So, the index was a value between 0 to 1.

**Trained provider.** The facility has a provider who has received training on routine care for labor and delivery in the past 2 years. **Kangaroo mother care:** The facility practices kangaroo mother care for low-birth-weight babies, a practice in low-income countries where incubator facilities are not readily available, especially at the lower level delivery care hospitals. **Guideline:** The facility has guidelines for either conducting a normal delivery or cesarean delivery. **Facility location- urban/ rural:** Although the study facilities are rural hospitals situated at the sub-district level, some UHCs are in semi-urban/ urban areas, mainly serving the urban population from the catchment urban areas. Irrespective of their location, every UHC covers a rural population, and this study will explore if the location or area makes a difference in the delivery service utilization.

**Partograph.** The facility uses a partograph to monitor labor and delivery. **Availability of midwife/ nurse-midwife:** The facility has at least one midwife/ nurse- trained in midwife to provide delivery service. Sub-district hospitals very often have vacant positions for midwives and midwifery nurses. If there was at least one midwife/ nurse trained in midwifery available on duty was considered as available midwife/ nurse midwife. **Total beds:** Usually UHCs are either 31-bed or 50-bed hospitals. However, there were a few 20-bed facilities, and one facility had 100 beds.

## Results

### Comparison of the service availability in 2014 and 2017

There is no significant difference in the number of facilities providing antenatal care, normal delivery, and cesarean delivery service between the 2014 and 2017 survey periods. Table 2 shows that only 31 facilities in 2014 and 32 facilities in 2017 provided both normal and cesarean delivery. Facilities practicing kangaroo mother care increased from 78 (57.8%) to 88 (64.7%) between the two survey periods. Ninety-one facilities (64.7%) conducted regular reviews of maternal and neonatal deaths or near misses in 2014 compared to 78 facilities (57.4%) in 2017.

In terms of service utilization, 29% of the facilities had 53-183 deliveries per year, another 29% of the facilities had 184-365 deliveries per year and only 26% facilities had 500 plus deliveries. Each facility, on average, had 588 admissions seeking maternal health care and conducted 385 delivery cases on average per year per facility. Each hospital had an average of 157 cases of intrapartum or delivery complications, and 48 of those complications were referred to a different hospital.

**Table 2. Availability of maternal health services in 2014 and 2017.**

| Maternal health services | 2014 | | 2017 | |
|---|---|---|---|---|
| | n (%) | Total facilities (N) | n (%) | Total facilities (N) |
| Ante natal care | 139 (99.3) | 140 | 140(99.3) | 141 |
| Normal delivery | 135 (96.4) | 140 | 136(96.5) | 141 |
| Caesarean delivery | 31 (22.1) | 140 | 32 (22.7) | 141 |
| Both caesarean and normal delivery | 31 (22.1) | 140 | 32 (22.7) | 141 |
| Post-natal care | -- | -- | 139(98.6) | 141 |
| Blood group and testing | 104 (74.3) | 140 | 103 (73) | 141 |
| Practice Kangaroo mother care | 78 (57.8) | 135 | 88 (64.7) | 136* |
| Conduct regular review of maternal or neonatal death or near misses | 91 (64.7) | 135 | 78 (57.4) | 136* |
| **Service utilization**[a] | | | | |
| Number of deliveries conducted in a year | | | | |
| *<52* | – | – | *10 (7.5)* | 134 |
| *53-183* | – | – | *38 (28.5)* | 134 |
| *184-365* | – | – | *39 (29.1)* | 134 |
| *366-500* | – | – | *12 (9.0)* | 134 |
| *501+ delivery* | – | – | *35 (26.1)* | 134 |
| Average delivery per facility | – | – | 384.54 (SD 390.61) | 134 |
| Average admission for maternity care per facility | – | – | 587.71 (SD 565.92) | 134 |
| Average complications per facility | – | – | 156.56 (SD 187.73) | 134 |
| Average referrals per facility | – | – | 47.85 (SD 72.17) | 134 |
| Average stillbirth | – | – | 8.82 (SD 11.08) | 134 |
| Average neonatal death | | | 0.81 (SD 2.71) | 134 |
| Average maternal death | | | 0.12 (SD 0.39) | 134 |

*statistically significant difference at α = 0.05 level; a Delivery utilization data is not available for the year 2014; SD-standard deviation.

Table 3 shows each indicator of nine signal function availability at a certain point of time in the facility. A total of 135 facilities in 2014 and 136 facilities in 2017 offered normal delivery service, and these facilities were asked during the survey if they have performed/offered any of the 9 services (signal function indicators) to their clients. However, two of those indicators, availability of c-section delivery and blood transfusion system, were asked to all 140 facilities in 2014 and 141 facilities in 2017.

The ability to administer parental antibiotics was quite the same in 2014 and 2017. However, anticonvulsant and oxytocic services increased. Assisted vaginal delivery service, another lifesaving service to treat obstructed delivery, was available in 113 facilities in 2014 and reduced to 84 facilities in 2017.

The average or mean facility readiness score was 0.67 in 2014 and became 0.69 in 2017. Only a few facilities had all 9 lifesaving services available in 2014 and 2017. Eight (6.1%) facilities in 2014 had all nine signal function indicators available compared to 11 (8.1%) facilities in 2017, and the differences were not statistically significant. Sixty-four facilities (48.5%) in 2014 had at least seven signal functions available, while 75 (55.6%) offered the same service in 2017 (Table 3).

Fig 1 shows the number of hospitals that performed EmONC signal function (SF) indicators in the past three months during 2014 and 2017. While slight improvements are observed

**Table 3. Number and percentage of facilities having five, seven and nine signal functions available.**

| Quality indicators | UHCs 2014 | | UHCs 2017 | |
|---|---|---|---|---|
| | N (%) | Total N | N (%) | Total N |
| Parenteral antibiotics | 121 (89.6) | 135 | 123 (90.4) | 136 |
| Parenteral oxytocic | 115 (85.2) | 135 | 129 (94.9) | 136 |
| Anticonvulsant | 78 (57.8) | 135 | 85 (62.5) | 136 |
| Assisted vaginal delivery | 113 (83.7) | 135 | 84 (61.8) | 136 |
| Manual removal of placenta | 111 (82.2) | 135 | 119 (87.5) | 136 |
| Removal of retained product of conception | 93 (68.9) | 135 | 98 (72.1) | 136 |
| Neonatal resuscitation | 105 (77.8) | 135 | 117 (86.0) | 136 |
| Caesarean delivery | 31 (22.1) | 140 | 32 (22.7) | 141 |
| Blood transfusion | 31 (22.1) | 140 | 54 (38.3) | 141 |
| **At least 5 indicators** | 102 (77.27) | 132 | 112 (82.96) | 135 |
| **At least 7 indicators** | 64 (48.5) | 132 | 75 (55.6) | 135 |
| **All 9 indicators** | 8 (6.1) | 132 | 11 (8.1) | 135 |
| **Facility Readiness** | 0.67 | 132 | 0.69 | 135 |

in some indicators, such as parenteral oxytocin (111 to 127), neonatal resuscitation (90 to 105), and blood transfusion (24 to 46), other key indicators, including cesarean delivery (29 to 30) and anticonvulsant use (72 to 69), show little to no progress. Additionally, the percentage of facilities achieving all 9 indicators remains very low, increasing only marginally from 6 to 11 facilities. Although there is a minor increase in the SF Index score from 0.62 in 2014 to 0.64 in 2017, the findings indicate that overall readiness did not significantly improve between the two survey years. Hence, conclusions regarding improvement should be interpreted with caution.

## Infrastructure

**Resource for providing delivery service.** Thirty-one facilities in 2014 and 32 facilities in 2017 provided cesarean delivery care (see Table 4). Human resource availability of these facilities was studied, and analysis showed that 23 facilities in 2014 had a surgeon available or on call 24 hours to provide cesarean delivery compared to 25 facilities in 2017. All 23 facilities providing cesarean care in 2014 had a schedule of work or call list for that provider compared to 15 facilities out of 25 facilities in 2017 had a call list in 2017. A surgeon and anesthetist, commonly known as a pair, are crucial for providing c-section delivery. In 2014, 22 facilities had a pair compared to 18 facilities in 2017. All 141 facilities surveyed in 2014 were asked if they had at least one midwife or nurse trained in midwifery. Only 48% of these facilities met this criterion. Facilities without a trained midwife or nurse could still conduct normal deliveries, but cases involving complications were typically referred to higher-level facilities.

*Functional cesarean delivery equipment.* Table 5 shows the list of cesarean delivery equipment available and functional in 2014 and 2017 among the facilities that provide cesarean delivery service. Only 27 facilities had a functional anesthesia machine in 2014 compared to 28 facilities in 2017. All 31 facilities in 2014 and 32 facilities in 2017 had operation tables, lights, and intra-vine stands.

## Empirical models for association

Table 6 presents the association between delivery rates and independent variables, including the facility readiness score, availability of trained providers for routine labor and delivery care, practice of kangaroo mother care, total number of beds, presence of delivery guidelines,

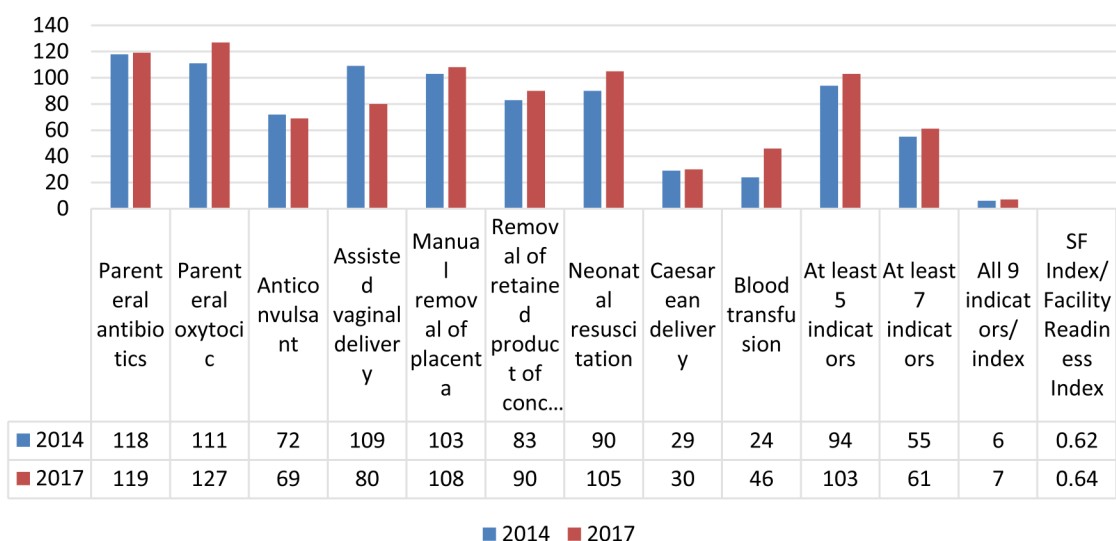

**Fig 1. Number of hospitals performed SF indicators in the past 3 months.**

availability of a midwife or nurse midwife, facility location, and use of partographs. Of the 140 facilities surveyed in 2017, 127 were included in the analysis, as others either did not provide delivery care or lacked delivery rate data.

The analysis found significant associations between delivery rates and facility readiness (p = 0.009), availability of trained providers (p = 0.004), and total beds (p = 0.008). A 0.01-unit increase in the readiness index was associated with 143.61 additional deliveries, and facilities with trained providers saw an increase of 54.46 deliveries. Cesarean delivery rates (Model 2) were also significantly associated with the facility readiness index (p = 0.003), total beds (p = 0.071), and facility location (p = 0.03). These findings emphasize the importance of facility readiness, infrastructure, and personnel availability in improving delivery service utilization.

## Discussion

This study assessed the readiness of sub-district hospitals in Bangladesh to deliver Emergency Obstetric and Newborn Care (EmONC) and found persistent gaps that limit service capacity. While there were modest improvements in specific signal functions, such as the availability of blood transfusion services and neonatal resuscitation, overall readiness showed little progress between 2014 and 2017. Most facilities failed to meet the required standards for comprehensive care, with only 8% providing all nine signal functions in 2017. The limited availability of cesarean delivery services, shortages of trained personnel, and inadequate supplies underscore significant barriers to scaling up comprehensive care. These findings reflect systemic challenges in ensuring equitable access to high-quality maternal and newborn care at the sub-district level.

Overall, 63% of facilities deliver fewer than 500 deliveries annually, which indicates that these facilities may not be a preferred destination for childbirth. The availability of signal function indicators was low, with only modest improvement between the two survey periods. This may also explain the lower delivery rates in the majority of facilities. In comparison, studies showed that only 2% of births per year in Finland, less than 5% of births in England, and less than 8% of births per year in the USA occurred in healthcare facilities with fewer than 500 deliveries per year [26–28].

**Table 4. Cesarean delivery surgeon and anesthetist availability.**

| Provider availability | 2014 | N (%) | 2017 | N (%) |
|---|---|---|---|---|
| Facility has a C-section provider present or on call 24 hours | 23 (74.2) | 31 | 25 (78.1) | 32 |
| Facility has duty schedule or call list of cesarean delivery provider for 24 hours | 23 (100) | 23 | 15 (60.0) | 25 |
| Facility has anesthetist present or on call 24 hours a day (including weekends and on public holidays | 22(71.0) | 31 | 20 (62.5) | 32 |
| Facility has duty schedule or call list of anesthetist's assignments for 24 hours | 20 (90.9) | 22 | 12 (60.0) | 20 |
| Facility has both C-section provider and anesthetist | 22 (71.0) | 31 | 18 (56.22) | 32 |
| Nurse midwife/midwife* | – | – | 66 (47.5) | 141 |

*All 141 facilities answered this question.

**Table 5. Cesarean delivery equipment availability.**

| Available and functioning | 2014 n (%) | 2017 n (%) |
|---|---|---|
| Anesthesia machine | 27 (87.1) | 28 (87.5) |
| Tubing and connector | 25 (80.6) | 25 (78.1) |
| Oropharyngeal airway (adult) | 27 (87.1) | 25 (78.1) |
| Oropharyngeal airway (pediatric) | 20 (64.5) | 15 (46.9) |
| Magills forceps- adult | 26 (83.9) | 28(87.5) |
| Magills forceps-pediatric | 24 (77.4) | 19 (59.4) |
| Endotracheal tube cuffed sizes 3.0 – 5.0 | 29 (93.5) | 24 (75.0) |
| Endotracheal tube cuffed sizes 5.5 – 9.0 | 27 (87.1) | 22(68.8) |
| Intubating stylet | 20 (64.5) | 21 (65.6) |
| Spinal needle | 28 (90.3) | 28 (87.5) |
| Operation table | 31 (100) | 32 (100) |
| Operation table light | 31 (100) | 32 (100) |
| Intra vine stand | 31 (100) | 32 (100) |
| Emergency power supply | 30 (96.8) | 30 (93.8) |
| Air conditioner | 27 (87.1) | 28 (87.5) |
| Oxygen cylinder with Flowi | 31 (100) | 30 (93.8) |
| Oxygen cylinder without Flowi | 0(0) | 28 (87.5) |
| N | 31 | 32 |

The facility readiness score was low both in 2014 and 2017 and remained unchanged between the two survey periods. The same was true for human resources and medical appliances and supplies. Only 11 out of 135 facilities had all nine SF indicators at any point in time in 2017 compared to 8 facilities in 2014. A study conducted in Bangladesh showed that SF indicators were available in district hospitals and maternal and child welfare centers but absent in sub-district hospitals [29]. The findings of less availability of signal functions are consistent with the previous study conducted in public and private hospitals in Bangladesh [30].

The analysis also showed that the SF availability index did not change significantly between 2014 and 2017. These findings are similar to individual studies showing weak infrastructure, staffing, and emergency care capacity in low-income countries [10,15,31–34]. No apparent change in the quality between the two surveys could be the possible reason why maternal

**Table 6. Regression model examining the association between facility delivery rates and facility readiness, represented by the Signal Function Index (SF Index), in 2017.**

| | Model 1: Delivery Rate (N = 127) | | Model 2: C-Section Delivery rate (N = 127) | |
|---|---|---|---|---|
| | Coefficient (β) | P value | Coefficient (β) | P value |
| Facility Readiness | 143.61 | **0.009** | 43.32 | **0.003** |
| Facility has a provider received training on routine care for labor and delivery in past 2 years (ref. no) | 54.46 | **0.004** | -0.013 | 0.998 |
| Facility practice kangaroo mother care (ref. no) | 15.83 | 0.540 | 5.194 | 0.443 |
| Total bed | 3.487 | **0.008** | 0.614 | 0.071 |
| Facility has guideline for delivery (ref. no) | 31.413 | 0.205 | 5.824 | 0.367 |
| Facility has delivery midwife/nurse midwife (ref. yes) | -23.162 | 0.329 | 1.005 | 0.871 |
| Urban/ rural (ref. yes) | -40.234 | 0.097 | -13.447 | **0.030** |
| Use partographs to monitor labor/delivery (ref. yes) | -4.48 | 0.86 | -6.722 | 0.311 |

death did not decrease. Bangladesh still has high maternal mortality and only 65% of facilities in 2014 and 57% of facilities in 2017 regularly reviewed maternal or neonatal deaths or near-missed cases. On average, 391 delivery care was provided per facility per year, causing 162 complications, and each facility referred 48 complicated cases in 2017. Maternal death at the facility was low (0.13 deaths per facility), but stillbirth, neonatal death, and near-missed cases were high in number. This big burden of complications, referrals, stillbirths, and neonatal deaths could have been avoided with a robust infrastructure and competent and alert staff. Emergency referral is a common practice for community-level or lower-level facilities to deal with complications. However, long distances to next-level facilities, road conditions, transport fares, and slow recognition of the severity of the complications by the client's attendants are barriers to reaching a referral hospital, resulting in maternal band newborn death during referral [35, 36].

We noted a significant association between the obstetric care readiness index and delivery rate. Additional factors associated with delivery rate are the providers having training on routine care for labor and delivery in the past 2 years, total bed number, and facility location, i.e., urban-rural. This finding aligns with Kruk and colleagues [15], who also noted a strong association between the facility readiness index and the co-variates- delivery rate, number of skilled staff per bed, facility type, i.e., public vs private, etc.

Our study brings important findings and adds to the scientific literature on readiness of care, identifying gaps in the provision of comprehensive obstetric care. Readiness improvement is a persistent challenge in most LMICs. Countries adopt various policies to improve utilization and readiness. For example, India took the 'Janani Suraksha Yojana' (JSY) program, which provides a cash incentive for women to deliver in facilities[37]. The JSY program increased facility delivery by 50% in the first year and, to some extent, reduced neonatal mortality, but no effect was found on maternal mortality [37–39]. Bangladesh introduced a similar program- the 'Demand Side Finance Voucher Program' in the hard-to-reach rural areas. The available evaluation found that Bangladesh's voucher program did not improve quality [40].

Many lower-income countries have experimented with promoting delivery in higher-rate settings [41, 42]. However, this approach means that the number of facilities with the capability to do cesarean sections might also need to increase and might not be feasible for Bangladesh in a short period of time. To meet the Sustainable Development Goals, Bangladesh must

carefully choose the policy and implement it effectively to meet its obligations to the population to provide safe delivery of care services.

We had several limitations. First, SPA data is collected using a cross-sectional survey method, not a longitudinal survey method. So, the comparison between 2014 and 2017 does not exactly reflect the quality changes among the same facilities. Secondly, we did not have the utilization/outcome data (number of deliveries, complications, referrals, etc.) for the year 2014, and therefore, we could not compare outcome data in two different survey periods. We had data on staffing at healthcare facilities, but we do not have data for provider skill or competence in obstetric care, especially. We also could not measure the respectful treatment during delivery as this element was out of our scope. Disrespectful behavior and abuse are common and might discourage women from seeking care. Future studies might bring a more comprehensive quality assessment and bring important insights; our index provides a checklist to proof if facilities possess cesarean delivery and complication management capacity.

Those limitations notwithstanding, to the best of our knowledge, this is the first study using nationally representative health system data combined with outcome/ utilization data to systematically assess the comprehensive obstetric care quality at two different time points. Further studies are needed to build on this research. Determinants of quality need to be investigated more.

## Conclusion

This study highlights critical gaps in the readiness of sub-district hospitals in Bangladesh to provide comprehensive Emergency Obstetric and Newborn Care (EmONC) services. While some improvements were observed in the availability of signal functions like oxytocin and anticonvulsants between 2014 and 2017, overall readiness remained low, with only 8% of facilities meeting all nine signal functions in 2017. Facilities with higher readiness indices reported significantly higher delivery rates, emphasizing the importance of facility readiness in improving service utilization.

To address these gaps, targeted efforts are needed to strengthen infrastructure, human resources, and the availability of essential supplies. Ensuring comprehensive EmONC services at sub-district hospitals is vital to reducing maternal and newborn mortality and achieving sustainable improvements in health outcomes across Bangladesh.

## Author contributions

**Conceptualization:** Kaji Korotki, M. Mahmud Khan.

**Formal analysis:** Kaji Korotki.

**Investigation:** Nabil Natafgi.

**Methodology:** Kaji Korotki, M. Mahmud Khan.

**Resources:** Kaji Korotki, Mark Macauda, Syed Abdul Hamid, M. Mahmud Khan.

**Software:** Kaji Korotki.

**Supervision:** Nabil Natafgi, Mark Macauda, Syed Abdul Hamid, M. Mahmud Khan.

**Validation:** Kaji Korotki, Nabil Natafgi, Mark Macauda, Syed Abdul Hamid, M. Mahmud Khan.

**Visualization:** Nabil Natafgi.

**Writing – original draft:** Kaji Korotki.

**Writing – review & editing:** Kaji Korotki, Nabil Natafgi.

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
