## [Decision Letter · Decision Letter 0]

4 Apr 2024

PONE-D-24-01445Association between ‘Quality of Care’ and Delivery Service Utilization in Bangladesh: Evidence from National Health Facility Assessment SurveysPLOS ONE

Dear Dr. Keya-Korotki,

Thank you for submitting your manuscript to PLOS ONE. After careful consideration, we feel that it has merit but does not fully meet PLOS ONE’s publication criteria as it currently stands. Therefore, we invite you to submit a revised version of the manuscript that addresses the points raised during the review process.

Dear authors,

After careful review, both I and another reviewer have evaluated your paper. We recommend revising the manuscript thoroughly for further development. Here are the major comments:

Introduction: It's suggested to combine the first two paragraphs into one cohesive problem statement. The first paragraph's first sentence should remain intact, while the remainder should be moved to the Methods section. Paragraph two should be split, with sentences 3-6 forming a new paragraph focusing on national strategies employed to reduce the MMR. Further elaboration on these strategies, such as upgrading facilities for Emergency Obstetric and Newborn Care (EmONC), promoting facility-based deliveries, and ensuring quality care, should be provided.Definition and metrics of QOC: There's concern that the definition and metrics of QOC might be misleading. Consider aligning them with established frameworks like Donabedian's Input, Process, and Outcome measures. Alternatively, if retaining the current definition, consider renaming it to reflect the concept more accurately, such as "EmONC facility readiness" or "EmONC performance."Temporal relations and multicollinearity: Address temporal relations and multicollinearity between EmONC functions and outcome measures, delivery and CS rates. Suggested approaches include using facility readiness of EmONC instead of actual performance as an independent variable, or redefining process metrics according to Donabedian quality metrics by excluding signal functions related to outcome measures like assisted delivery, CS, and blood transfusion.Choice of datasets: Clarify the rationale behind using the 2014 and 2017 datasets and specify which dataset is utilized for regression analysis.

We believe these revisions will enhance the quality and clarity of your manuscript.

Best regards,

We look forward to receiving your revised manuscript.

Kind regards,

Gizachew Tadele Tiruneh, Ph.D.

Academic Editor

PLOS ONE

Journal Requirements:

4. We noticed you have some minor occurrence of overlapping text with the following previous publication(s), which needs to be addressed:

https://journals.plos.org/plosone/article?id=10.1371%2Fjournal.pone.0187238

https://www.thelancet.com/journals/langlo/article/PIIS2214-109X(16)30180-2/fulltext

In your revision ensure you cite all your sources (including your own works), and quote or rephrase any duplicated text outside the methods section. Further consideration is dependent on these concerns being addressed.

6. Please include a caption for figure 1.

7. Please upload a copy of Figure 1, to which you refer in your text on page 21 in PDF submission. If the figure is no longer to be included as part of the submission please remove all reference to it within the text.

8. Please include your tables as part of your main manuscript and remove the individual files. Please note that supplementary tables (should remain/ be uploaded) as separate ""supporting information"" files

**Additional Editor Comments:**

Dear authors,

I and another reviewer have evaluated your paper. We recommend revising the manuscript thoroughly for further development. Here are the major comments:

1) Introduction: It's suggested to combine the first two paragraphs into one cohesive problem statement. The first paragraph's first sentence should remain intact, while the remainder should be moved to the Methods section. Paragraph two should be split, with sentences 3-6 forming a new paragraph focusing on national strategies employed to reduce the MMR. Further elaboration on these strategies, such as upgrading facilities for Emergency Obstetric and Newborn Care (EmONC), promoting facility-based deliveries, and ensuring quality care, should be provided.

2) Definition and metrics of QOC: There's concern that the definition and metrics of QOC might be misleading. Consider aligning them with established frameworks like Donabedian's Input, Process, and Outcome measures. Alternatively, if retaining the current definition, consider renaming it to reflect the concept more accurately, such as "EmONC facility readiness" or "EmONC performance."

3) Temporal relations and multicollinearity: Address temporal relations and multicollinearity between EmONC functions and outcome measures, delivery and CS rates. Suggested approaches include using facility readiness of EmONC instead of actual performance as an independent variable, or redefining process metrics according to Donabedian quality metrics by excluding signal functions related to outcome measures like assisted delivery, CS, and blood transfusion.

4) Choice of datasets: Clarify the rationale behind using the 2014 and 2017 datasets and specify which dataset is utilized for regression analysis.

We believe these revisions will enhance the quality and clarity of your manuscript.

Best regards,

Reviewers' comments:

Reviewer's Responses to Questions

**Comments to the Author**

1. Is the manuscript technically sound, and do the data support the conclusions?

Reviewer #1: Partly

2. Has the statistical analysis been performed appropriately and rigorously? 

Reviewer #1: No

3. Have the authors made all data underlying the findings in their manuscript fully available?

Reviewer #1: Yes

4. Is the manuscript presented in an intelligible fashion and written in standard English?

Reviewer #1: No

5. Review Comments to the Author

Reviewer #1: Reviewer comments

The authors have made commendable effort to make sense of available secondary data to show EmONC QoC status in Bangladesh.

The manuscript requires major revisions

1. The title is too broad and misleading. First as the study is focused on EmONC which is an intervention designed to improve maternal and newborn health, the title should reflect it as such.

2. Abstract section

Background/introduction is missing

Objectives do not reflect the findings of the study

Methods : Indicate the number of surveyed health facilities in 2014 and 2017 and those providing basic EmONC and Comprehensive

Results :

o One of the objectives of your study is to assess service utilization but not shown under the result section

o Present the figures in proportion/percentage in addition to the #, to make it more informative

o Please stratify the findings by basic and comprehensive EmONC

o Conclusion : Without showing the proportion the given information/data are not enough to conclude that there is improvement. You need to show whether there were statistically significant change in the part you presented the results to claim that there are improvement in 2017 compared to 2017.

The conclusion has gone far from what you have shown under the results

3. Background/Introduction section

• It lacks logical flow

• It requires high level of analysis and removing redundant ideas

• The objective of the manuscripts you have it here is different from the one in the abstract section. It should be reconciled. The one you have it here reflect your study better.

4. Methods section

Please make it clear which facilities are providing basic EmONC and comprehensive EmONC. Without having this information, it is difficult to review the paper. Please clearly describe the standards for Emergency Obstetrics and Neonatal Care (EmONC) recommended by WHO and contextualization of the Bangladeshi system . Eg

EmONC standard indicators Basic EmoNC facility Comprehensive EmONC Facility

Population served 25,000? 100,000?

Availability per district

Signal functions 7 9

Health care workers

Supplies and pharmaceuticals

Infrastructure

# of facilities surveyed in 2014 (SPA)

# of facilities surveyed in 2017 (SPA

In the SPA survey, how many of the 140 and 141 facilities surveyed in 2014 and 2017 respectively were basic EmONC, comprehensive EmONC and non-EmONC (Be remined those facilities not providing either the 7 or the 9 signal functions are non-EmONC, which may be excluded from the analysis.

All the analysis should be stratified by Basic and comprehensive. If you wish to present only the comprehensive EmONC, you can do so but clearly indicate in your objective, methods and background.

Although EmoNC has clear standard on the type of infrastructure, supplies, medicine, manpower and services provided, your manuscript gave emphasis on C. Section and required resources for it. 1) It is not clear why only C. Section was given emphasis in the first place, 2) by focusing on C. Section, the manuscript could limit its relevance for few area experts.

Are the health facilities surveyed in 2014 and 2017 the same? Are they like cohorts? Often there are some facilities to be added or removed when having cross-sectional SPA survey in different places.

It is not clear what do you mean by “the 2017 SPA survey was conducted in 2019”. Did you mean the 2017 SPA reports were published, in 2019 ?

As you described, Donabedian is a comprehensive model that enable to assess structure, process and outcomes. However, in your study when calculating the QoC index you looked into the availability of signal functions, (only the process). By doing so you diverged from the model assumption/premises, which is not clear. Make it clear why you chose this model and how it guided your study. I would suggest you to have a good understanding of the model, QoC and the EmONC concepts and applications, then it would be easy for you to be guided by the model for assessing EmONC care quality.

The description under the delivery service utilization sub-title should be described under variable and measurement section not under the analysis. Moreover, the facility categorization you are referring to is made to show how repeated exposure/practice increase skill, care quality and hence health outcomes. If you are applying this, you should also categorize the facilities/SPA data accordingly based on the volume of care they provide taking into account the national standards.

In EmONC, the signal function related to neonatal care is “ Perform basic neonatal resuscitation”. You have Kangaroo Mother Care (KMC), which is not EmONC signal function but can be an additional variable but you should have a variable to measure a signal function for newborn care as recommended by the EmONC standard.

On the part about the availability of midwife/nurse, indicate how you quantified/ classified as per the EmONC standard under variables and measurement section eg; for the # of deliveries the facility conducts how many midwives/nurses are there. Are they below or above the expected standard as these would have significant implication on the QoC and to show the health facility readiness?

Result section

You have some information in the results section, which should be part of the method section. Eg: about the analyzed samples, " Out of 141 facilities in 2017, a total of 7 facilities had missing information and finally 134 facilities (141-7=134 facilities) were included in the service utilization analysis"

Why death review is highlighted in the first paragraph of all the maternal and newborn health indicators? There is no mention in the background or method section related to it.

When comparing 2014 with 2017 data and other comparisons, test of significance is important to show whether there are changes between the two survey points.

In table 2, you have listed all the nine-signal function for comprehensive EmONC. From the figure, I can see few of the facilities (around one fourth of the surveyed facilities were providing comprehensive EmONC services). Here, the analysis for the availability of C. Section and blood transfusion should be applied only for comprehensive EmONC facilities as you did it in table 3 and 4.

The application of QoC index and EmONC standard lacks integration. For example, a basic EmONC facilities providing all the seven signal functions should have a QoC index of 100%. But in your manuscript, it can be 77% (i.e seven out of the 9 signal functions).

On page 14 just before the caption for table 3, "All 141 facilities in 2014 were asked if they had a nurse midwife or midwife and only 48% of these facilities had a midwife/ nurse" my questions are 1) are you referring to a single nurse or midwife? You might need to give more contextual information, if any SPA surveyed health facilities can be run without a nurse or a midwife. Eg. Low level dispensaries may not need to have nurse/midwife. Taking this in mind your assessment should be focused on facilities providing EmONC services. 2) is there any health facility providing delivery are without having a midwife or a nurse to the least?

On page 15, just before the caption for table 5 "Analysis found a strong association between delivery rate and QoC index (p value 0.009), availability of trained provider (p value 0.004), total bed (p value 0.008) and the facility location (p value 0.097). We found that when QoC index increased by 0.01-unit, delivery rate increased by 143.61 delivery. Similarly, if facility has a provider trained on routine care for labor and delivery, delivery rate increased by 54.46 delivery and the association is statistically significant with a p value of 0.004" Here I see the chicken and the egg phenomenon. In the background/methods you have given evidence that facilities having more delivery have better quality of care. Hence is it the # of service provided which brings quality or is it the QoC which attracted more clients to come to the facility? which is difficult to answer using data from a cross sectional survey. Longitudinal design such as cohort could have been appropriate to address. Therefore, you should be careful in interpreting the findings, you can show association but not causal link/causality using the SPA and DHS data.

Discussion section

On the first paragraph of the discussion section, …" Only 22% facilities (31 out of 140) in 2014 and 23% facilities (32 in 141 facilities) in 2017 had cesarean delivery service". It is misleading to say only 22% /23% of the facilities in Bangladesh provide C-section. According to the EmONC standard only about one fourth/fifth of all health facilities are expected to provide C-section service, hence the prevalence should be calculated out of those supposedly providing the services. I would suggest you to re-run the analysis with the appropriate denominator

On page 16 paragraph 3 it says "The quality of delivery care was low both in 2014 and 2017 and remained unchanged between the two survey periods. Only 11 out of 135 facilities had all nine SF indicators at any point of time in 2017 compared to 8 facilities in 2014" It seems that you have used quality of delivery care to mean quality of EmONC by analyzing the availability of the nine-signal functions as predictor. I disagree as the two are different. In the analysis, you have included all the facilities irrespective of their EmONC status. But you should consider only the comprehensive EmONC facilities to assess whether they perform the nine signal functions.

General comments

The manuscript would benefit from language revision.

Here is a link for a paper on BEmONC status from Ethiopia that used primary and secondary data. https://bmcpregnancychildbirth.biomedcentral.com/articles/10.1186/1471-2393-14-354#Tab4

I have attached my comments as word document and track-changed manuscript

6. PLOS authors have the option to publish the peer review history of their article (what does this mean?). If published, this will include your full peer review and any attached files.

Reviewer #1: No

---

## [Author Response · Author response to Decision Letter 1]

17 Nov 2024

November 17, 2024

Gizachew Tadele Tiruneh, Ph.D.

Academic Editor

PLOS ONE

Dear Dr. Tiruneh,

Thank you for the opportunity to address the comments of peer-reviewers who have reviewed our manuscript, “Association between ‘emergency obstetric & newborn care readiness’ and Delivery Service Utilization in Bangladesh: Evidence from National Health Facility Assessment Surveys” (Submission ID PONE-D-24-01445R). In this letter, we explain how we have responded to the suggestions made by the reviewers, with reference to specific edits in the manuscript text as necessary. We are submitting a “clean” revised manuscript and the recommend revised version with tracked changes as a supplementary file.

Editor

1. Introduction: It's suggested to combine the first two paragraphs into one cohesive problem statement. The first paragraph's first sentence should remain intact, while the remainder should be moved to the Methods section. Paragraph two should be split, with sentences 3-6 forming a new paragraph focusing on national strategies employed to reduce the MMR. Further elaboration on these strategies, such as upgrading facilities for Emergency Obstetric and Newborn Care (EmONC), promoting facility-based deliveries, and ensuring quality care, should be provided.

Response:

Thank you for the suggestion. We have combined the first two paragraphs into a single cohesive problem statement. Relevant details from the Introduction have been relocated to the Methods section. Additionally, a new paragraph elaborating on national strategies to reduce the maternal mortality ratio (MMR), including upgrading EmONC facilities, promoting facility-based deliveries, and ensuring quality care, has been added.

2. Definition and metrics of QOC: There's concern that the definition and metrics of QOC might be misleading. Consider aligning them with established frameworks like Donabedian's Input, Process, and Outcome measures. Alternatively, if retaining the current definition, consider renaming it to reflect the concept more accurately, such as "EmONC facility readiness" or "EmONC performance."

Response:

We agree that the QOC is misleading and has been a matter of discussion internally amongst authors. Accordingly, we have replaced the term "Quality of Care (QoC)" with "Facility Readiness" to better reflect the scope of the study.

3. Address temporal relations and multicollinearity between EmONC functions and outcome measures, delivery and CS rates. Suggested approaches include using facility readiness of EmONC instead of actual performance as an independent variable, or redefining process metrics according to Donabedian quality metrics by excluding signal functions related to outcome measures like assisted delivery, CS, and blood transfusion.

Response:

We have adopted the suggestion to use facility readiness as the independent variable. The alternative approach was not feasible, as excluding signal functions related to outcome measures would reduce the sample size significantly (from 127 facilities to 32), limiting the robustness of the analysis.

4. Clarify the rationale behind using the 2014 and 2017 datasets and specify which dataset is utilized for regression analysis.

Response:

The 2014 and 2017 datasets were selected because they are the two most recent available surveys. The regression analysis was conducted using the 2017 dataset. Due to the COVID-19 pandemic, the survey scheduled for 2020 was postponed, making the 2017 dataset the latest available for analysis. This was clarified in the methods section.

Reviewer 1

1. The title is too broad and misleading. First, as the study is focused on EmONC, which is an intervention designed to improve maternal and newborn health, the title should reflect it as such.

Response:

The title has been revised to accurately reflect the study's emphasis on EmONC readiness and its association with delivery service utilization.

2. Abstract section

Background/introduction is missing

Objectives do not reflect the findings of the study

Methods : Indicate the number of surveyed health facilities in 2014 and 2017 and those providing basic EmONC and Comprehensive

Results :

o One of the objectives of your study is to assess service utilization but not shown under the result section

o Present the figures in proportion/percentage in addition to the #, to make it more informative

o Please stratify the findings by basic and comprehensive EmONC

o Conclusion : Without showing the proportion the given information/data are not enough to conclude that there is improvement. You need to show whether there were statistically significant change in the part you presented the results to claim that there are improvement in 2017 compared to 2017.

The conclusion has gone far from what you have shown under the results

Response:

Background information has been added to the abstract. Objectives have been revised for alignment with the findings. Details regarding the number of surveyed facilities in 2014 and 2017, as well as their provision of basic and comprehensive EmONC, have been clarified.

We assessed overall facility readiness instead of stratifying the analysis for the following reasons:

1. The 2014 survey did not include a variable to distinguish between basic and comprehensive EmONC facilities. In the 2017 survey, bed numbers were used as a proxy, identifying 86 facilities with 50 beds (comprehensive), 46 with 31 beds (basic), and 3 with 10 beds (basic). Since both types of facilities serve similar population ranges (250,000–400,000), stratification would add limited value to the analysis.

2. Both basic and comprehensive facilities are the primary destinations for childbirth. Assessing overall readiness across two time periods provides a more holistic understanding of readiness trends.

3. Staffing shortages in sub-district hospitals diminish distinctions between basic and comprehensive facilities. Medical providers, nurse midwives, and technologists are frequently absent in rural areas, despite being officially posted to these facilities. Without adequate staffing, even comprehensive facilities may function at a basic level. For this reason, an overall assessment of readiness was conducted.

The proportion was added to the result section and the conclusion section was revised to better reflect the reality of the results: “While certain indicators improved, overall readiness stagnated due to shortages of anesthetists, gynecologists, and essential supplies. With 64% of surveyed sub-district hospitals classified as comprehensive care facilities, resource and staffing investments are crucial to enhance readiness and reduce maternal and newborn mortality.”

3. Background/Introduction section

• It lacks logical flow

• It requires high level of analysis and removing redundant ideas

• The objective of the manuscripts you have it here is different from the one in the abstract section. It should be reconciled. The one you have it here reflect your study better.

Response:

• To improve logical flow, we restructured the Background/Introduction section into cohesive paragraphs. This included revising transitions and grouping related information to create a clear narrative.

• We removed redundant ideas and added a more in-depth analysis of the context, particularly focusing on the importance of facility readiness for Emergency Obstetric and Newborn Care (EmONC) in addressing maternal and newborn mortality.

• The objectives in the manuscript and abstract have been reconciled to ensure consistency. The objectives from the manuscript, which more accurately reflect the study's focus, were adopted and incorporated into the abstract.

Methods section:

4. Please make it clear which facilities are providing basic EmONC and comprehensive EmONC. Without having this information, it is difficult to review the paper. Please clearly describe the standards for Emergency Obstetrics and Neonatal Care (EmONC) recommended by WHO and contextualization of the Bangladeshi system.

Response:

The manuscript has been revised to provide clear definitions and standards for Basic and Comprehensive EmONC as recommended by WHO and contextualized for Bangladesh. A table has been added (Table 1) to illustrate the population served, availability per sub-district, required signal functions, healthcare workers, supplies and pharmaceuticals, and infrastructure.

The 2014 survey did not include a variable to differentiate Basic and Comprehensive EmONC. However, the 2017 survey used "bed numbers" as a proxy, identifying 86 comprehensive (50 beds) and 46 basic (31 beds) facilities. This narrative is included in the revised Methods section to address this limitation.

Table 1. Characteristics and Standards of Basic and Comprehensive Emergency Obstetric and Newborn Care (EmONC) Facilities in Bangladesh

EmONC standard indicators Basic EmoNC facility Comprehensive EmONC Facility

Population served 250,000-400,000 250,000-400,000

Availability per sub-district 1 1

Signal functions 7 9

Health care workers Midwives or nurses trained in midwifery Obstetrician, anesthetist, midwives or nurses trained in midwifery

Supplies and pharmaceuticals Basic obstetric drugs (e.g., oxytocin, magnesium sulfate), basic instruments for vaginal delivery Full set of obstetric drugs, equipment for cesarean sections, and blood transfusion supplies

Infrastructure 31 beds 50 beds

# of facilities surveyed in 2014 (SPA) Not available Not available

# of facilities surveyed in 2017 (SPA 46 basic facilities 86 comprehensive facilities

5. All the analysis should be stratified by Basic and Comprehensive. If you wish to present only the Comprehensive EmONC, you can do so but clearly indicate in your objective, methods, and background.

Response

Stratification was considered but could not be implemented for the 2014 data due to the lack of a variable distinguishing Basic and Comprehensive EmONC. For 2017, the narrative explains that facilities were categorized based on bed numbers. However, the analysis was conducted using overall readiness to ensure consistency across both years and to avoid losing generalizability. The study objectives, methods, and background have been revised.

6. Although EmoNC has clear standard on the type of infrastructure, supplies, medicine, manpower and services provided, your manuscript gave emphasis on C. Section and required resources for it. 1) It is not clear why only C. Section was given emphasis in the first place, 2) by focusing on C. Section, the manuscript could limit its relevance for few area experts.

Response

The focus on cesarean sections was driven by its critical importance in addressing complicated deliveries, which significantly contribute to maternal mortality. Two thirds (64%) of the studied facilities in 2017 survey comprehensive EmONC. Complicated delivery cases, mostly, come to the hospitals since Bangladesh has a common practice of home birth and 50% of the deliveries take place at home. So the readiness of comprehensive signal functions was important. However, the manuscript has been revised to provide a balanced emphasis on all signal functions, including neonatal resuscitation and other maternal care indicators, to enhance its relevance for a broader audience.

7. Are the health facilities surveyed in 2014 and 2017 the same? Are they like cohorts?

Response:

The facilities surveyed in 2014 and 2017 are not identical cohorts. Some overlap exists, but the surveys were conducted cross-sectionally at different time points, with variations in included facilities. This clarification has been added to the Methods section.

8. It is not clear what you mean by 'the 2017 SPA survey was conducted in 2019'.

Response:

Thanks for pointing this out. The phrasing has been revised for clarity. The 2017 SPA survey, initially scheduled for 2017, was delayed due to logistical issues and was conducted in 2019. The revised manuscript explicitly mentions this delay and its context.

9. As you described, Donabedian is a comprehensive model that enable to assess structure, process and outcomes. However, in your study when calculating the QoC index you looked into the availability of signal functions, (only the process). By doing so you diverged from the model assumption/premises, which is not clear. Make it clear why you chose this model and how it guided your study. I would suggest you to have a good understanding of the model, QoC and the EmONC concepts and applications, then it would be easy for you to be guided by the model for assessing EmONC care quality.

Response:

We thank the reviewers for this comment. The Donabedian model was chosen for its comprehensive framework encompassing structure, process, and outcomes. In the context of this study, quality of care is operationalized as facility readiness, with the following components: Structure: Refers to the availability of logistics, supplies, equipment, and personnel (e.g., midwives, obstetricians, anesthetists). Process: Denotes the activities involved in providing care, such as performing the nine signal functions for Emergency Obstetric and Newborn Care (EmONC), including diagnosis, prescription, and the execution of critical interventions. Outcome: Refers to the results of care delivery, such as rates of facility-based deliveries, cesarean sections, and maternal or neonatal health outcomes.

While process indicators were emphasized, the model guided the interpretation of findings to identify actionable weaknesses in health systems. This rationale has been clarified in the revised manuscript.

10. The description under the delivery service utilization sub-title should be described under variable and measurement section not under the analysis. Moreover, the facility categorization you are referring to is made to show how repeated exposure/practice increase skill, care quality and hence health outcomes. If you are applying this, you should also categorize the facilities/SPA data accordingly based on the volume of care they provide taking into account the national standards.

Response:

The description of delivery service utilization has been relocated to the "Variables and Measurement" section for better alignment with the manuscript’s structure.

11. In EmONC, the signal function related to neonatal care is “ Perform basic neonatal resuscitation”. You have Kangaroo Mother Care (KMC), which is not EmONC signal function but can be an additional variable but you should have a variable to measure a signal function for newborn care as recommended by the EmONC standard.

Response:

The manuscript has been updated to include "basic neonatal resuscitation" as the signal function for newborn care, as per EmONC standards. Kangaroo Mother Care (KMC) has been retained as an additional variable relevant to LMIC contexts where incubator services are limited.

12. On the part about the availability of midwife/nurse, indicate how you quantified/ classified as per the EmONC standard under variables and measurement section eg; for the # of deliveries the facility conducts how many midwives/nurses are there. Are they below or above the expected standard as these would have significant implication on the QoC and to show the health facility readiness?

Response:

The availability of midwives/nurses was quantified based on the presence of at least one trained midwife or nurse on duty during the survey. This is consistent with the EmONC standard. The revised Methods section explicitly details this classification.

Result section

13. You have some information in the results section, which should be part of the method section. Eg: about the analyzed samples, " Out of 141 facilities in 2017, a total of 7 facilities had missing information and finally 134 facilities (141-7=134 facilities) were included in the service utilization analysis".

Response:

We agree with the reviewer and have moved this information to the methods section. The revised methods section now includes the sample details for clarity and proper alignment with the study structure.

14. Why death review is highlighted in the first paragraph of all the maternal and newborn health indicators? There is no mention in the background or method section related to it.

Response:

We have addressed this by incorporating maternal and neonatal death revi

---

## [Editor Report · Decision Letter 1]

3 Dec 2024

PONE-D-24-01445R1Association between ‘emergency obstetric & newborn care readiness’ and Delivery Service Utilization in Bangladesh: Evidence from National Health Facility Assessment SurveysPLOS ONE

Dear Dr. Natafgi,

Thank you for submitting your manuscript to PLOS ONE. After careful consideration, we feel that it has merit but does not fully meet PLOS ONE’s publication criteria as it currently stands. Therefore, we invite you to submit a revised version of the manuscript that addresses the points raised during the review process.

Thank you for addressing all the comments and significantly improving the manuscript. It reads much more clearly now, and I am confident it will make a valuable contribution to the literature.

A few additional comments;

Move the "Public Health Care Structure in Bangladesh" section to the Study Settings section.The study objectives are mentioned twice: once in the "Public Health Care Structure in Bangladesh" section and again in the "Introduction" section. Please combine them into one.Table 6: In the analysis section, it is reported that a multivariable regression analysis was conducted, but only unadjusted regression findings are presented here. Please include the adjusted coefficients as well. Additionally, add other structural measures from the SPA survey, such as the availability of basic amenities, equipment, facility managers, etc.Discussion: Begin by summarizing the key findings in the first paragraph, then interpret and discuss these findings in subsequent paragraphs. Avoid reporting results in the Discussion section. Focus more on interpretation and analysis.Conclusion: The conclusion section currently reads like a discussion. Please focus on summarizing the key findings and highlighting the main takeaways.Minor edits: edit the phrase “‘emergency obstetric & newborn care readiness’” under the title of the manuscript as title case “Emergency Obstetric and Newborn Care Readiness”; under Table 2, change the word “still birth” into “stillbirth”

We look forward to receiving your revised manuscript.

Kind regards,

Gizachew Tadele Tiruneh, Ph.D.

Academic Editor

PLOS ONE

Journal Requirements:

Additional Editor Comments:

Thank you for addressing all the comments and significantly improving the manuscript. It reads much more clearly now, and I am confident it will make a valuable contribution to the literature.

A few additional comments;

1. Move the "Public Health Care Structure in Bangladesh" section to the Study Settings section.

2. The study objectives are mentioned twice: once in the "Public Health Care Structure in Bangladesh" section and again in the "Introduction" section. Please combine them into one.

3. Table 6: In the analysis section, it is reported that a multivariable regression analysis was conducted, but only unadjusted regression findings are presented here. Please include the adjusted coefficients as well. Additionally, add other structural measures from the SPA survey, such as the availability of basic amenities, equipment, facility managers, etc.

4. Discussion: Begin by summarizing the key findings in the first paragraph, then interpret and discuss these findings in subsequent paragraphs. Avoid reporting results in the Discussion section. Focus more on interpretation and analysis.

5. Conclusion: The conclusion section currently reads like a discussion. Please focus on summarizing the key findings and highlighting the main takeaways.

6. Minor edits: edit the phrase “‘emergency obstetric & newborn care readiness’” under the title of the manuscript as title case “Emergency Obstetric and Newborn Care Readiness”; under Table 2, change the word “still birth” into “stillbirth”

---

## [Author Response · Author response to Decision Letter 2]

27 Dec 2024

December 27, 2024

Gizachew Tadele Tiruneh, Ph.D.

Academic Editor

PLOS ONE

Subject: Resubmission of Revised Manuscript Submission ID PONE-D-24-01445R1

Dear Dr. Tiruneh,

Thank you for the opportunity to submit a second revision of our manuscript titled "Association between ‘Emergency Obstetric & Newborn Care Readiness’ and Delivery Service Utilization in Bangladesh: Evidence from National Health Facility Assessment Survey". We appreciate the constructive feedback provided by you and the reviewers, which has greatly improved the quality of our work. In this revision, we have addressed all remaining comments, ensuring the manuscript meets the expectations of the journal.

Editor

1. Move the "Public Health Care Structure in Bangladesh" section to the Study Settings section.

Response:

Thank you for the suggestion. The "Public Health Care Structure in Bangladesh" section has been relocated to the Study Settings section for improved organization and flow.

2. The study objectives are mentioned twice: once in the "Public Health Care Structure in Bangladesh" section and again in the "Introduction" section. Please combine them into one.

Response:

The study objectives have been combined into a single, clear statement in the Introduction section.

3. Table 6: In the analysis section, it is reported that a multivariable regression analysis was conducted, but only unadjusted regression findings are presented here. Please include the adjusted coefficients as well. Additionally, add other structural measures from the SPA survey, such as the availability of basic amenities, equipment, facility managers, etc.

Response:

We appreciate the Editor’s observation regarding the presentation of regression coefficients in Table 6. To address any confusion, we have clarified the table heading by replacing "Unstandardized Coefficient (β)" with "Coefficient (β)." The coefficients presented in the table represent adjusted coefficients, as they account for the influence of all covariates included in the regression models. This ensures that the reported associations are controlled for potential confounders.

Additionally, standardized coefficients have been provided alongside adjusted coefficients (in the table below but not revised manuscript) for the Editor’s reference. Standardized coefficients allow for direct comparison of the relative importance of different variables, as they standardize the scale of all variables. While the standardized coefficients have not been included in the final manuscript, we are happy to incorporate them into the paper if the editor or reviewer feels they would provide further clarity.

Regarding the inclusion of additional structural measures, such as basic amenities, equipment, and facility management, the variables in the regression models were carefully selected based on theoretical frameworks, conceptual relevance, and findings from bivariate analyses. Key structural and process-related variables, such as facility readiness (measured by the QoC index), provider training, availability of midwives/nurse midwives, total beds, and the use of partographs, were prioritized as they directly impact delivery care. Other important aspects, such as life-saving medicines (antibiotics, oxytocin, anticonvulsants) and neonatal resuscitation, are captured within the readiness index. While facility managers play an indirect role in delivery care through supervision, direct care variables were prioritized for inclusion in the models.

We hope this explanation clarifies both the presentation of adjusted coefficients and the rationale for variable selection. Please let us know if further adjustments or additions are required.

4. Discussion: Begin by summarizing the key findings in the first paragraph, then interpret and discuss these findings in subsequent paragraphs. Avoid reporting results in the Discussion section. Focus more on interpretation and analysis.

Response:

Thank you for your suggestion. The Discussion section has been revised to begin with a summary of the key findings in the first paragraph. Subsequent paragraphs now focus on interpreting and analyzing these findings in detail, avoiding the repetition of results. This adjustment enhances the clarity and depth of the discussion.

5. Conclusion: The conclusion section currently reads like a discussion. Please focus on summarizing the key findings and highlighting the main takeaways.

Response:

The Conclusion section has been rewritten to focus solely on summarizing the key findings and main takeaways. Interpretative elements that belonged in the Discussion section have been relocated there.

6. Minor edits: edit the phrase “‘emergency obstetric & newborn care readiness’” under the title of the manuscript as title case “Emergency Obstetric and Newborn Care Readiness”; under Table 2, change the word “still birth” into “stillbirth”

Response:

Both edits have been corrected.

We believe these changes fully address the reviewers’ concerns and further strengthen the manuscript. Thank you for your continued consideration of our work. Please do not hesitate to contact us should you have any additional questions or require further clarifications.

---

## [Editor Report · Decision Letter 2]

30 Dec 2024

Association between ‘Emergency Obstetric & Newborn Care Readiness’ and Delivery Service Utilization in Bangladesh: Evidence from National Health Facility Assessment Surveys

PONE-D-24-01445R2

Dear Dr. Natafgi,

We’re pleased to inform you that your manuscript has been judged scientifically suitable for publication and will be formally accepted for publication once it meets all outstanding technical requirements.

Kind regards,

Gizachew Tadele Tiruneh, Ph.D.

Academic Editor

PLOS ONE

Additional Editor Comments (optional):

The authors have adequately addressed all comments, resulting in significant improvements to the manuscript. 
---

## [Editor Report · Acceptance letter]

PONE-D-24-01445R2

PLOS ONE

Dear Dr. Natafgi,

I'm pleased to inform you that your manuscript has been deemed suitable for publication in PLOS ONE. Congratulations! Your manuscript is now being handed over to our production team.

Kind regards,

on behalf of

Dr. Gizachew Tadele Tiruneh

Academic Editor

PLOS ONE